# Quality of Low-Carbon Steel as a Distribution of Pollution and Fatigue Strength Heated in Oxygen Converter

**Tomasz Lipiński** 

Faculty of Technical Sciences, University of Warmia and Mazury in Olsztyn, 10-719 Olsztyn, Poland; tomekl@uwm.edu.pl

**Abstract:** The works available in the literature presenting the influence of impurities on the properties (mainly fatigue strength) of material give an answer with a high degree of probability for hard steels and large precipitations (usually above 10 μm). The impact of non-metallic impurities on the durability of high-ductility steels causes much greater problems and is much more difficult to explain. The results of the existing studies rarely take into account the diameter of the impurities in relation to the distance between the impurities. This paper presents the results of tests carried out on a low-carbon steel heated in a 100-tonne oxygen converter and deoxidized under vacuum. The fatigue strength test was carried out on cylindrical samples using rotational bending for different tempering temperatures of the steel. The quotient of the average size of the inclusions and the average distance between the inclusions were analyzed. Based on the obtained results, it was found that steel annealed in the converter and vacuum degassed has a content of both phosphorus and sulfur below 0.02% and a total volume of impurities of 0.086%. The main fraction of impurities are oxide inclusions with a diameter below 2 μm. An increase in fatigue strength was found along with an increase in the number of impurities, mainly of small diameters.

**Keywords:** steel; low-carbon steel; fatigue strength; inclusions; impurities; distribution of impurities

## 1. Introduction

Steel is the main material used for the production of machine parts, devices and various types of constructions [1–3]. An important feature that distinguishes individual steel grades from each other is their quality [4,5]. The notion of quality is a very broad concept. There are a number of standards and guidelines defining the quality parameters of materials. In general, quality can be defined as meeting the expectations of the recipients, who are usually constructors at the stage of material selection. In industrial production, materials with different properties and qualities are needed. Different requirements are placed on materials intended for low-responsibility structures, and different requirements are placed on materials intended for elements of responsible structures operating under variable loads. In engineering applications, high-quality steels are called steels with high performance properties. The high purity of the material is essential to ensure these properties [6–10].

The steel production process carried out in industrial conditions requires the addition of pig iron and steel scrap to the furnaces. This material may contain additional impurities—e.g., in the form of non-metallic particles pressed in during the operation of the material—which are difficult to eliminate in the metallurgical process. Non-metallic inclusions can also infiltrate the liquid alloy in other ways. They can, among others, come off from the lining of furnaces, ladles, etc.; they can be impurities introduced from the outside into the furnace; but they can also arise as a result of metallurgical processes [11–14]. Many authors have described impurities in steel over the decades [15–19]. The conducted research constantly provides new data, thanks to which it is possible to rationally control the quality of steel produced to meet the specific, ever-higher requirements of customers. Research on the contents of impurities in steel and their impact on individual performance

properties [20–24], despite the large number of works in this field, are and will continue to be necessary for the continuous expansion of knowledge and the improvement of material and production processes. Results are required, among others, from the continuous improvement of production processes and increasing the requirements for steels as a result of the development of technology [25–28]. Among the many types of steel, low-carbon steel is very popular [29,30]. In the literature, there are a number of works presenting tests of its properties both at the stage of its production [31–37], during operation and in failure tests (premature damage) [38–42].

There are many works on improving the properties of construction materials. These works present, among others, the results of research on the surface treatment of materials [43,44] using computer simulations [45–47] and fatigue tests [48–51]. The research results indicate that the quality of a metal alloy is determined by its chemical composition, microstructure and, in the next stage, purity [52,53]. There are also known methods of increasing the purity of steel by melting it under synthetic slag [54]. Good results were obtained as a result of steel production using secondary metallurgy [55–57] and in converter processes [58]. Steel degassing in a vacuum is a process that allows up to 70% oxygen reduction after using degassing during tapping [59]. The analysis of the works presented above shows that degassing in a vacuum reduces the oxygen content in steel, which reduces the possibility of the formation of large-diameter oxide impurities. Confirmation of the above requires further research to determine the morphology and distribution of inclusions in steel, and then their impact on the quality expressed by one of the more stringent indicators, which is fatigue strength.

A large amount of research is carried out on steels produced in laboratory conditions. These conditions, despite the fact that they enable the production of material with the assumed parameters, do not reflect the actual state. Research conducted on an industrial scale is therefore of great importance. The research available in the literature carried out in real steel rarely describe the size of non-metallic inclusions, as well as the distances between them. The influence of inclusions on the fatigue strength (one of the most sensitive parameters due to the influence of impurities), as is described in many works, still does not explain the clear role of impurities present in steel [60–63]. It is known that large brittle non-metallic inclusions occurring in materials with high hardness and low plasticity reduce the fatigue strength [64–68]. This effect is more visible the closer impurities with a large diameter are to the surface of the working element [69,70]. The problem becomes less obvious with high quality and, therefore high purity, ductile steels (having few impurities) [2,71,72]. These steel grades should now be the main source of interest for researchers. The above-mentioned shortcomings prompted the author to undertake this subject of research.

The main purpose of the work was to determine the pollution distribution index as the quotient of the impurity diameter and the distance between the impurities for particular dimensional ranges of impurities in high-quality ductile steel produced through the converter process subjected to vacuum degassing, and to determine the impact of non-metallic inclusions on the fatigue strength.

## 2. Materials and Methods

In order to increase the credibility of the research results, the decision was made to conduct the research in an industrial production process. The steel was melted in a 100-tonne oxygen converter and deoxidized under vacuum. The results obtained from 6 melts were used for the analysis. The melts were carried out with the use of steel scrap in the amount of 26–28% by the weight of the entire charge. The method of continuous casting with degassing in a vacuum was used. Then, $100 \times 100$ mm square billets were rolled using conventional methods. In each of the melts, samples were taken from the ingots for testing. The chemical composition was determined using the AFL quantometer and LECO analyzers; conventional methods were also used. The relative total volume of non-metallic inclusions was determined using the extraction method. The impurity field

with a diameter of 2 μm was measured using a Quantimet video inspection microscope at 400× magnification. Software constraints were used to determine impurities, allowing the measurement of inclusions that were: greater than 2 μm, greater than 5 μm, greater than 10 μm. The analysis was carried out assuming that the quotient of the surface of the impurities and the area of the measurement surface are equal to the quotient of the volume of particles and the analyzed volume. The use of the automatic analysis method for inclusions with dimensions below 2 μm gave quite large errors. For this reason, the volume of impurities smaller than 2 μm was calculated from the difference between the total volume of impurities obtained from the chemical method and the volume of impurities with a diameter greater than 2 μm. The measurements of the diameter of non-metallic inclusions for each melt were carried out using a video microscope on 10 samples and using classical methods on 7 samples.

The real average chemical composition of the tested steel and its standard deviation from six heats carried out in the oxygen converter is presented in Table 1.

**Table 1.** Average chemical composition and its standard deviation of low-carbon steel from 6 heats.

| Chemical Element | C | Mn | Cr | Ni | Mo | Si | Cu | P | S | B |
|---|---|---|---|---|---|---|---|---|---|---|
| | | | | | | **wt. %** | | | | |
| Contents | 0.24 | 1.18 | 0.52 | 0.52 | 0.24 | 0.24 | 0.02 | 0.017 | 0.015 | 0.003 |
| Standard deviation | 0.02 | 0.06 | 0.03 | 0.03 | 0.02 | 0.02 | 0.005 | 0.002 | 0.003 | 0.0005 |

Cylindrical samples with a diameter of 10 mm intended for fatigue strength testing were taken from billets. The longitudinal axes of the samples were parallel to the longitudinal axis of the rolled billets. The test samples were prepared in accordance with the PN-H/74-04327:1974 standard [56]. Because they constantly work in different conditions, a different microstructure is produced in them, which determines different mechanical properties. In order to diversify the microstructure and mechanical properties of the tested steel, it was subjected to hardening and various variants of tempering. The austenitizing temperature was set at 880 °C. The tempering temperatures were set in the range between 200 °C and 600 °C, varying them every 100 °C. Bending fatigue strength was performed on a VEB Werkstoffpruf-maschinen rotary bending machine. The tests were carried out in accordance with the PN-74/H-04327 standard [73]. The fatigue strength test was carried out until failure or until 107 cycles were reached. An experimentally determined load was used to guarantee an average durability of $10^4$ cycles, depending on the steel tempering temperature: for 200 °C—650 MPa, for 300 to 500 °C—600 MPa and for 600 °C—540 MPa. The results of the tests carried out on 318 samples from 6 independent melts were used to analyze the fatigue strength for rotational bending. The analysis of individual parameters (both concerning inclusions and strength) was carried out on the arithmetic average, calculated separately for each melt.

Pollution distribution index $\alpha$, calculated as the quotient of the average size of pollutants and the average distance between pollutants, is shown in formula (1). The average distance between impurities $\lambda$ was calculated from formula (2) [74]. The pollutant distribution index $\alpha$, as a function of the contents of pollutants with different diameters, was presented in the form of regression equations defined by the formula (3). The bending fatigue strength of the tested steel after hardening from 880 °C and tempering for the assumed tempering temperatures is presented in the general form (4):

$$\alpha = \frac{\overline{d}}{\overline{\lambda}} \qquad (1)$$

$$\overline{\lambda} = \frac{2}{3}\overline{d}\left(\frac{1}{V} - 1\right) \qquad (2)$$

$$\alpha_i = a \cdot V_i + b, \tag{3}$$

$$z_{go((tempering\ temperature))} = a \cdot V + b, \tag{4}$$

where:

$\overline{d}$—average diameter of impurity, μm,

$\overline{\lambda}$—arithmetic average distance between impurities, μm,

$a$, $b$—coefficients of the equation,

$V$—relative volume of impurities, %,

$V_i$—relative volume of impurities for the assumed its diameter, %.

The statistical significance of regression Equations (3) and (4) was checked on the basis of the Student's $t$ distribution function for $\alpha = 0.05$ and degrees of freedom f = $n - 1$ using formula (5).

$$t = \frac{r}{\sqrt{\frac{1-r^2}{n-1}}} \tag{5}$$

where:

$r$—correlation coefficient,

$n$—number of measurements.

The results of measuring the volume of the impurities obtained during the research were verified experimentally for selected heats. For this purpose, non-metallic inclusions were selected from the unit volume of steel by means of chemical extraction, which were tested by determining the size of individual inclusions using a computer method for the same diameter ranges for which the surface analysis was carried out. The difference for non-metallic inclusions with a diameter greater than 2 μm did not statistically exceed 5% by volume.

## 3. Results

The microstructures of the steel hardened at 880 °C and the different tempering temperatures, between 200 °C and 600 °C, steel are presented in Figure 1.

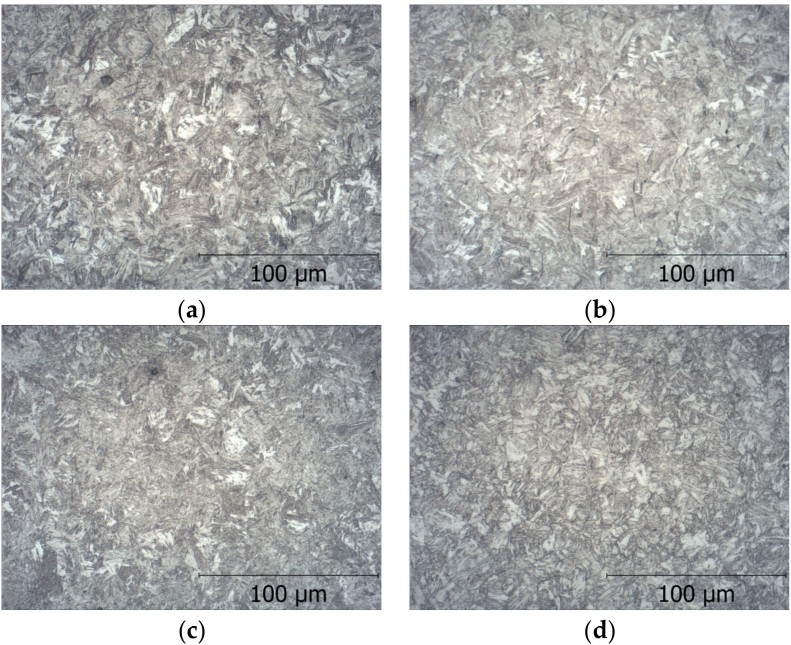

**Figure 1.** *Cont.*

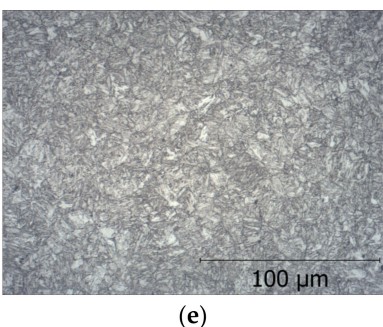

(**e**)

**Figure 1.** Microstructure of the steel hardened at 880 °C and tempered at: (**a**) 200 °C—low tempered martensite, (**b**) 300 °C—medium tempered martensite, (**c**) 400 °C medium tempered martensite, (**d**) 500 °C—high tempered martensite, and (**e**) 600 °C—high tempered martensite, etch. nital.

Examples of non-metallic inclusions in the tested steel are shown in Figure 2 [75].

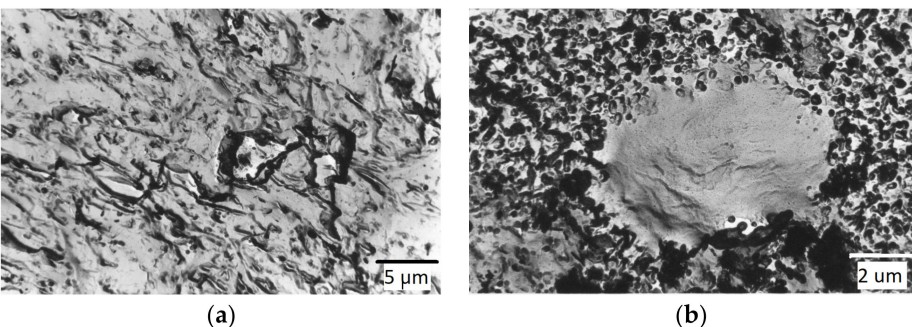

(**a**)             (**b**)

**Figure 2.** Non-metallic inclusions: (**a**) $CaO \cdot Al_2O_3$, (**b**) $SiO_2 \cdot MnO \cdot FeO$.

The percentage relative volume of impurities as a function of its diameter is presented in Figure 3 for: V—total contents of impurities, $V_0$—contents of impurities with a diameter of less than 2 μm, $V_1$—contents of impurities with a diameter larger than 2 μm, $V_2$—contents of impurities with a diameter between 2 and 5 μm, $V_3$—contents of impurities with a diameter larger than 5 μm.

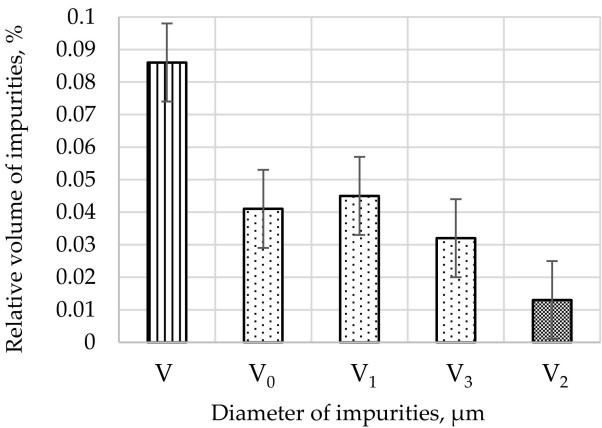

**Figure 3.** Percentage relative volume of impurities as a function of its diameter: V—total contents of impurities, $V_0$—contents of impurities with a diameter of less than 2 μm, $V_1$—contents of impurities with a diameter larger than 2 μm, $V_2$—contents of impurities with a diameter between 2 and 5 μm, $V_3$—contents of impurities with a diameter larger than 5 μm.

The pollution distribution index $\alpha$ as a function of the contents of the total contents of impurities V is shown in Figure 4.

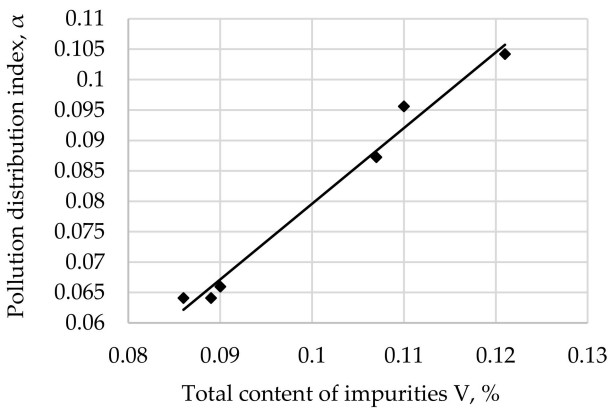

**Figure 4.** Pollution distribution index $\alpha$ as a function of the contents of total contents of impurities, V.

The pollution distribution index $\alpha$ as a function of the contents of impurities with a diameter of less than 2 μm $V_0$ is shown in Figure 5.

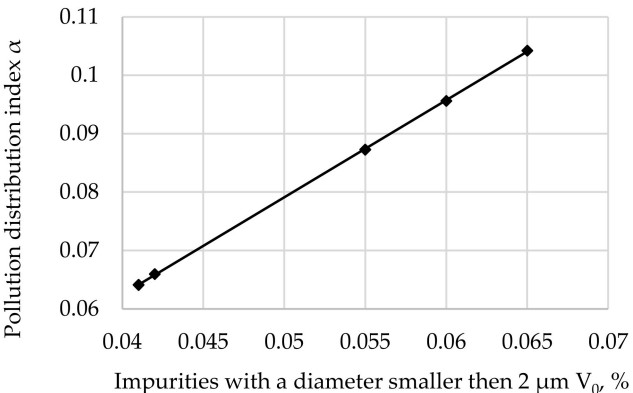

**Figure 5.** Pollution distribution index $\alpha$ as a function of the contents of impurities with a diameter of less than 2 μm, $V_0$.

The pollution distribution index $\alpha$ as a function of the contents of impurities with a diameter larger than 2 μm $V_1$ is shown in Figure 6.

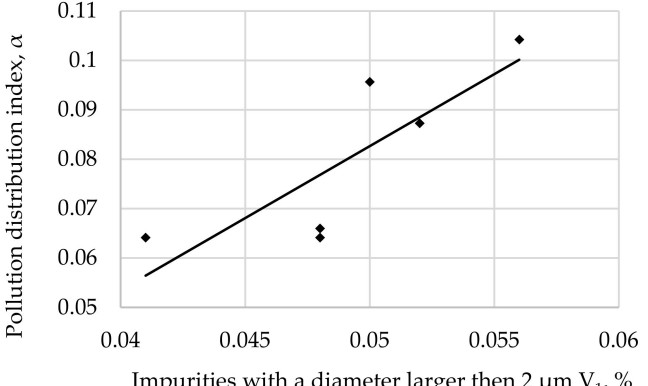

**Figure 6.** Pollution distribution index $\alpha$ as a function of the contents of impurities with a diameter larger than 2 μm, $V_1$.

The pollution distribution index $\alpha$ of impurities as a function of the contents of impurities with a diameter between 2 and 5 μm $V_2$ is shown in Figure 7.

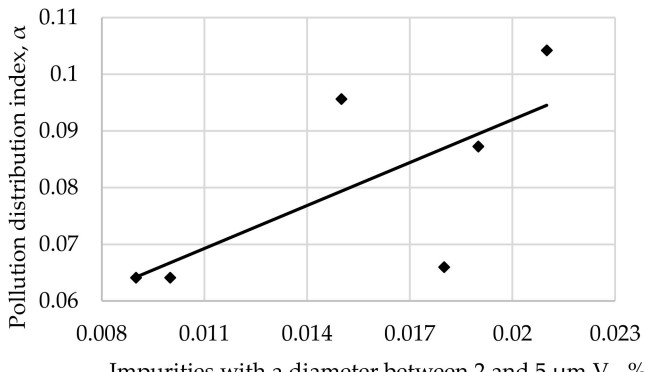

**Figure 7.** Pollution distribution index $\alpha$ of impurities as a function of the contents of impurities with a diameter between 2 and 5 μm, $V_2$.

The pollution distribution index $\alpha$ of impurities as a function of the contents of impurities with a diameter larger than 5 μm $V_3$ is shown in Figure 8.

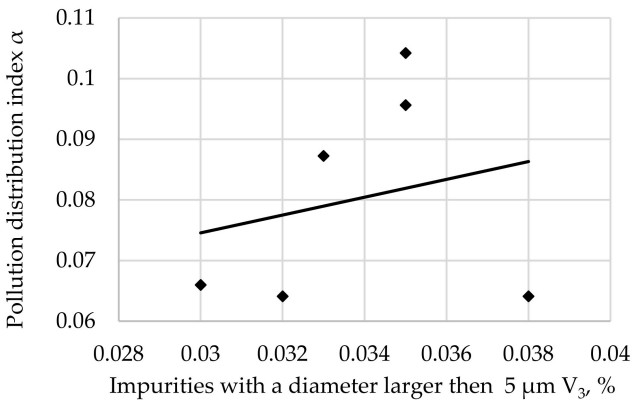

**Figure 8.** Pollution distribution index $\alpha$ of impurities as a function of the contents of impurities with a diameter larger than 5 μm, $V_3$.

The bending fatigue strength of low-carbon steel after hardening from 880 °C and tempering at 200 °C depending on the total contents of impurities V is shown in Figure 9.

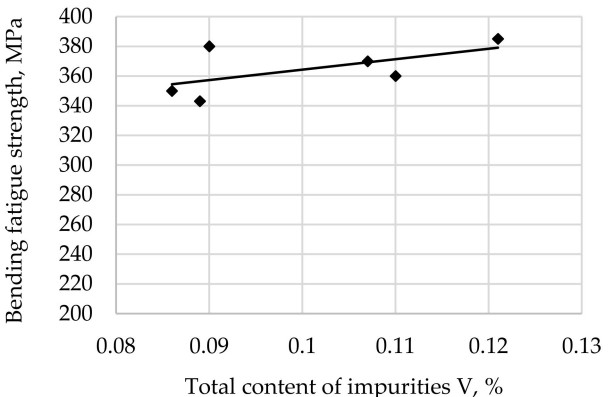

**Figure 9.** Bending fatigue strength of low-carbon steel after hardening from 880 °C and tempering at 200 °C depends on total contents of impurities V.

The bending fatigue strength of low-carbon steel after hardening from 880 °C and tempering at 300 °C depending on the total contents of impurities V is shown in Figure 10.

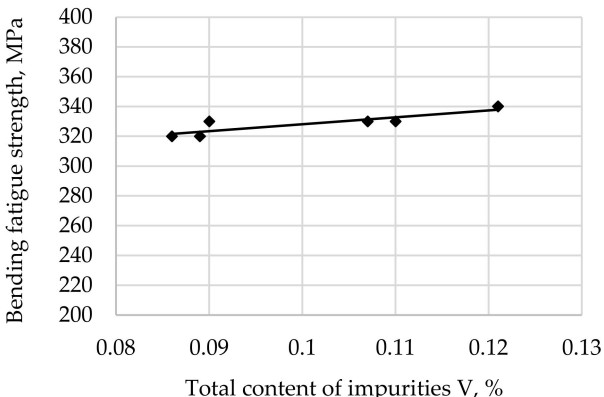

**Figure 10.** Bending fatigue strength of low-carbon steel after hardening from 880 °C and tempering at 300 °C depends on total contents of impurities V.

The bending fatigue strength of low-carbon steel after hardening from 880 °C and tempering at 400 °C depending on the total contents of impurities V is shown in Figure 11.

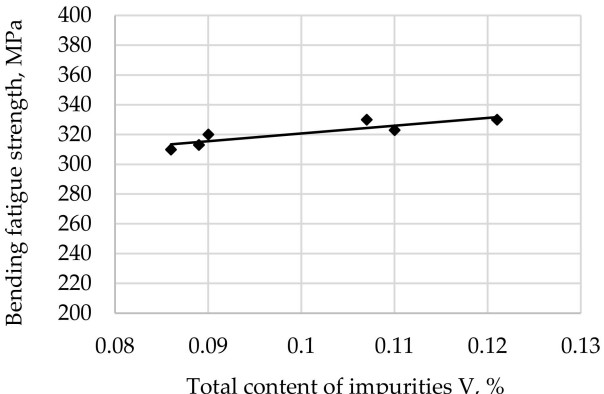

**Figure 11.** Bending fatigue strength of low-carbon steel after hardening from 880 °C and tempering at 400 °C depends on total contents of impurities V.

The bending fatigue strength of low-carbon steel after hardening from 880 °C and tempering at 500 °C depending on the total contents of impurities V is shown in Figure 12.

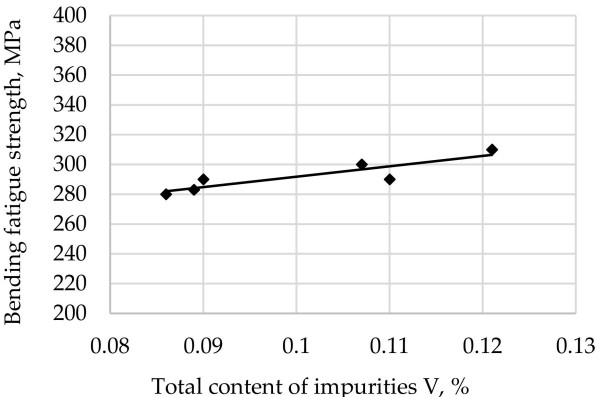

**Figure 12.** Bending fatigue strength of low-carbon steel after hardening from 880 °C and tempering at 500 °C depends on total contents of impurities V.

The bending fatigue strength of low-carbon steel after hardening from 880 °C and tempering at 600 °C depending on the total contents of impurities V is shown in Figure 13.

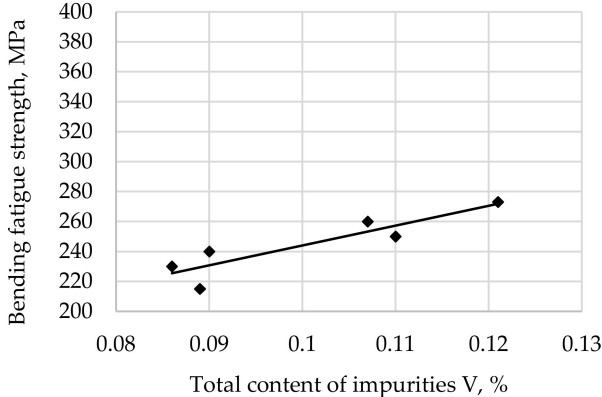

**Figure 13.** Bending fatigue strength of low-carbon steel after hardening from 880 °C and tempering at 600 °C depends on total contents of impurities V.

The statistical parameters representing pollutant distribution index $\alpha$ (3) are summarized in Table 2.

**Table 2.** Statistical parameters representing pollutant distribution index $\alpha$.

| Diameter of Impurities, μm | Regression Coefficient a (3) | Regression Coefficient b (3) | Correlation Coefficient r | $t_{\alpha = 0.05}$ Calculated by (5) | $t_{\alpha = 0.05}$ from Student's Distribution for $p = (n - 1)$ |
|---|---|---|---|---|---|
| All diameters | 1.2455 | −0.045 | 0.9922 | 17.798 | |
| Smaller then 2 μm | 1.6624 | −0.004 | 0.9999 | 158.102 | |
| Larger then 2 μm | 2.9360 | −0.063 | 0.8178 | 3.178 | 2.571 |
| Between 2 and 5 μm | 2.5269 | 0.0415 | 0.6996 | 2.189 | |
| Larger then 5 μm | 1.4707 | 0.0305 | 0.2302 | 0.529 | |

The statistical parameters representing the bending fatigue strength of the tested steel after hardening from 880 °C and tempering for the assumed tempering temperatures are presented in Table 3.

**Table 3.** Statistical parameters representing bending fatigue strength.

| Tempering Temperature °C | Regression Coefficient a (4) | Regression Coefficient b (4) | Correlation Coefficient r | $t_{\alpha = 0.05}$ Calculated by (5) | $t_{\alpha = 0.05}$ from Student's Distribution for $p = (n - 1)$ |
|---|---|---|---|---|---|
| 200 | 704.13 | 293.90 | 0.6003 | 1.678 | |
| 300 | 462.46 | 281.86 | 0.8712 | 3.968 | |
| 400 | 521.13 | 268.63 | 0.8808 | 4.160 | 2.571 |
| 500 | 696.67 | 222.15 | 0.8866 | 4.286 | |
| 600 | 132.70 | 111.33 | 0.9003 | 4.625 | |

## 4. Discussion of the Research Results

The microstructure of the steel after hardening and low tempering at 200 °C is composed of low-tempered martensite. Such a microstructure has a high hardness (average 429 HV), Figure 1a. After medium tempering in the range of 300 °C, as a result of the precipitation of carbides $\varepsilon$ and the subsequent disappearance of the supersaturation of martensite, a regular martensite microstructure with very fine carbides of submicroscopic dimensions was obtained, with a hardness lower than before, at the level of 389 HV (Figure 1b). After tempering at 400 °C, a mixture of ferrite and very fine cementite particles was obtained,

creating a troostite microstructure with a hardness of 379 HV (Figure 1c). After tempering at 500 °C, a mixture of very fine spherical cementite particles in a ferrite matrix was obtained, forming a sorbitol microstructure with a hardness of 338 HV. After increasing the tempering temperature to 600 °C, the cementite particles coagulated—which was associated with the growth of some cementite particles and the dissolution of small particles—a slight growth of the previously obtained sorbitol phase was obtained. For this temperature, the average hardness was 264 HV. As a result of hardening to martensite and tempering at different temperatures, the microstructure was differentiated from low-tempered martensite, through medium-tempered martensite, to high-tempered martensite. The change in the microstructure caused by the carbon diffusion during tempering resulted in a decrease in the hardness of the steel, and thus to changes in its mechanical properties.

In the tested steel, the occurrence of both non-metallic inclusions in the form of compounds, e.g., $Al_2O_3$, as well as complex inclusions, e.g., $CaO\ Al_2O_3$, $SiO_2\ MnO\ FeO$, are described in the literature [76–78].

The average carbon content in the tested steel is 0.24%. This allows this steel to be classified as low-carbon steel. The contents of phosphorus and sulfur are low and did not exceed 0.02% (Table 1). The percentage relative volume of impurities was set at 0.086% (Figure 3), which confirms the high quality of the steel. The contents of impurities with a diameter of less than 2 μm were set at 0.041%, and the contents of impurities with a diameter larger than 2 μm were set at 0.045%. The contents of impurities with a diameter larger than 5 μm were set at 0.032%. The contents of impurities with a diameter between 2 and 5 μm were determined to be 0.013% (Figure 3). Taking into account the fact that non-metallic inclusions with a diameter of less than 2 μm, with their small unit volume, occupy 0.041% of the volume of steel, and 48% of the volume of impurities of all dimensions, it was found that they constitute the largest group. It should therefore be expected that with a very small number of inclusions with a diameter greater than 5 μm, inclusions with small diameters will have a decisive influence on the properties of the analyzed steel.

The pollution distribution index $\alpha$ for all diameters (the entire dimensional range of inclusions) is a linear function directly proportional to the share of impurities present in the steel (Figure 4). Similarly, for a diameter smaller than 2 μm (Figure 5), the highest fit of the regression function was obtained with a correlation coefficient of 0.99. For impurities with a diameter greater than 2 μm, a greater dispersion of results was observed (lower correlation coefficient r = 0.82), Figure 6. Nevertheless, the obtained relationship is still statistically significant (Table 2). For larger impurity diameters, the equations were not statistically significant at the significance level of 0.05 (Table 2) (regression coefficients 0.70, Figure 7 and 0.23, Figure 8, respectively). The presented results prove the high reliability of the tests for small particles with a diameter of 2 μm. For particles with a larger diameter, the results are less reliable. It should be assumed that the low reliability of the results for impurities above 5 μm results from their small amount, taking into account the total share of impurities in the volume of steel. Using the pollution distribution index $\alpha$, it is possible to describe, with high accuracy, the total share of pollutants or very fine pollution with a diameter of less than 2 μm. Analyzing the pollution distribution index $\alpha$ for individual diameters of non-metallic inclusions (Figures 4–8), an increase in the pollution distribution index was found with the increase in the number of impurities of all dimensions present in the steel. The very high pollution distribution index correlation coefficient for impurities with a diameter of up to 2 μm confirms that inclusions with these diameters are the main inclusions in the analyzed steel. Each successive of the analyzed dimensional ranges, representing inclusions with increasingly large diameters, is characterized by greater dispersion, and thus a lower correlation coefficient (Table 2). Thus, the cumulative effect of non-metallic inclusions on steel properties will be mainly represented by fine inclusions.

For the bending fatigue strength tempered at 200 °C depending on the total content of impurities V (Figure 9), a regression coefficient of 0.60 was obtained and the result is statistically insignificant (Table 3). For the remaining tempering temperatures, the regression equations describe changes in the bending fatigue strength depending on the

total content of impurities V in a statistically significant way (Table 3). An increase in the correlation coefficients of the regression equations was noted with the increasing tempering temperature (Figures 10–13). This proves the increasing fit of the line described by the equation to the measurement points, and therefore the higher reliability of the results. Based on these results, it can be assumed that the non-metallic inclusions in the matrix with higher plasticity affect the fatigue strength in a more stable (predictable) way. It was also confirmed that the fatigue strength decreased with the increase in the tempering temperature, which is obvious, and can be explained by the decrease in the strength of the steel with the increase in the tempering temperature. For all of the tempering temperatures, an increase in the bending fatigue strength was noted with an increase in the share of impurities in the steel microstructure; the higher the tempering temperature, the more intense the increase. This is evidenced by the coefficient a in Equation (4), Table 3. One can risk a statement that for high purity steel, with a very small number of inclusions with a diameter above 5 μm, and at the same time with high steel ductility, non-metallic inclusions do not cause a decrease in the fatigue strength of the material, but rather, cause it to grow. The greater the plasticity of the steel (obtained as a result of the higher tempering temperature), the greater the increase. Most likely, the stresses occurring in the steel can then relax at the boundaries of the inclusions. This can lead to the so-called wave effect. The above effects may affect the increase in the bending fatigue strength along with the increase in the amount of fine non-metallic inclusions, which, according to Figure 1, constitute the main fraction of impurities. This assumption seems to be consistent with the results [79]. Non-metallic inclusions with a diameter exceeding 5 μm account for 0.013% (Figure 1) of the volume in steel. Assuming their random distribution in the steel microstructure and the fact that their impact on the steel properties will be proportional to their share in the microstructure [10,17,38,64], it is highly likely that the statistically low impact of these large impurities on the fatigue strength can be expected. However, it may be the case for products with a small diameter, e.g., chain links, that inclusions of large diameters occur in one of the links, weakening the cross-section, which may lead to accelerated wear of the element.

## 5. Conclusions

The presented test results showed that the non-metallic inclusions that occur in high-ductility steel do not necessarily cause the deterioration of its bending fatigue stress. Thus, it does not confirm the typical views presented in the literature, mainly concerning hard and low-plastic steels.

The steel heated in the converter and vacuum degassed has a very high purity, which is reflected in the content of both phosphorus and sulfur, below 0.02%, and the total volume of impurities of 0.086%.

The pollution distribution index α describes the morphology of all impurities present in high-purity steel very well, as well as very fine impurities with a diameter of up to 2 μm.

The main fraction of impurities present in the tested steel are oxide inclusions with a diameter below 2 μm.

A greater dependence of the bending fatigue strength was found with the increasing tempering temperature.

For steel with high purity and a very low content of inclusions with a diameter above 5 μm (0.013% volume of steel), an increase in the fatigue strength was found with an increase in the number of impurities with a very fine dimensional structure.

The steel produced in the converter and vacuum degassed is supposed to reduce the sensitivity of the fatigue life to stress oscillations, causing decohesion.

Comparing the fatigue properties with the dimensional structure of impurities, it can be assumed that submicroscopic inclusions (with a diameter of less than 2 μm) in plastic steels constitute barriers hindering the movement of dislocations. In addition, by absorbing the energy that causes the formation of discontinuities, they slow down the decohesion process.

Due to the high fineness of the inclusions in the tested steel, as well as the pronounced effect of the analyzed impurities on the fatigue strength, studies to elucidate the effect of very fine particles (with a diameter of less than 2 μm) on the fatigue life of steel seem interesting.

**Funding:** This research received no external funding.

**Institutional Review Board Statement:** Not applicable.

**Informed Consent Statement:** Not applicable.

**Data Availability Statement:** Not applicable.

**Conflicts of Interest:** The authors declare no conflict of interest.

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
