# Peer review of "Quality of Low-Carbon Steel as a Distribution of Pollution and Fatigue Strength Heated in Oxygen Converter"

_coatings, doi:10.3390/coatings13071275_

Round 1
Reviewer 1 Report
1. The abstract should present in a concise manner the main outcome of the paper. The current abstract should be polished along this line. Please bear in mind that a sharp abstract is central for article visibility. The abstract also lacks the support of key experimental data.
2. The major problem of this study is the lack of relevant microstructure images, which are very important to support the correctness of the study results. Additional relevant microstructure photographs are recommended.
3. The relevant parameters in formula 5 are missing descriptions.
4. In Figure 1, what is the relationship between the different size of impurities, whether V=V0+V1+V2+V3.
5. The test method section is too rough. Please give details of the conditions of the fatigue test.
6. The effects of impurities on fatigue crack formation and propagation need to be discussed in depth, and additional micrographs of fatigue fractures and microstructures in different heat treatment states are also recommended.
7. The conclusion should summarize the main findings and clarify how the results presented in the paper have made the field progressing. Future perspectives and research directions should also be included. A sole summary of the findings is not sufficient. Please polish the conclusion along this view.
8. This paper is like a statistical report, and many key results are not given. It need to be seriously revised.
Please carefully go through the manuscript to polish the language, including abstract.
Author Response
Dear Reviver,
Thank you for your constructive review
- The abstract should present in a concise manner the main outcome of the paper. The current abstract should be polished along this line. Please bear in mind that a sharp abstract is central for article visibility. The abstract also lacks the support of key experimental data.
Abstract changed.
- The major problem of this study is the lack of relevant microstructure images, which are very important to support the correctness of the study results. Additional relevant microstructure photographs are recommended.
The steel microstructure for individual tempering temperatures is shown in new Fig. 1.
- The relevant parameters in formula 5 are missing descriptions.
Supplemented descriptions of symbols in (5)
- In Figure 1, what is the relationship between the different size of impurities, whether V=V0+V1+V2+V3.
It's written in the manuscript:
„V – total contents of impurities,
V0 – contents of impurities with a diameter of less than 2 µm,
V1 – contents of impurities with a diameter larger than 2 µm,
V2 – contents of impurities with a diameter between 2 and 5 µm,
V3 - contents of impurities with a di-ameter larger than 5 µm.”
then
V≠V0+V1+V2+V3;
V=V0+V1;
V=V0+V2+V3.
- The test method section is too rough. Please give details of the conditions of the fatigue test.
The test conditions have been suplemented. I don't know what else should be added. The rest is at standards.
- The effects of impurities on fatigue crack formation and propagation need to be discussed in depth, and additional micrographs of fatigue fractures and microstructures in different heat treatment states are also recommended.
The microstructure of the steel for individual tempering temperatures is shown in Fig. 1. The request to supplement the microstructures of fatigue cracks is difficult to meet due to the adopted test method. Fatigue testing of hard steels, and thus those characterized by high brittleness, the test results of which are willingly presented, can be very easily discussed and illustrated. As can be seen in the literature, these steels break suddenly, leaving typical fatigue crack lines and a good fractographic pattern. You can consider the breakthrough zone at will. You can even stop the sample and break the sample, such cases are also described by the authors of the works. The steel used for the tests is a new construction material with a carbon content of 0.25% and, importantly, the addition of boron. Such steel has considerable ductility after hardening and tempering. It doesn't break violently. Its fatigue occurs gradually, and the traces of the original cracking are gradually obliterated as a result of the ongoing rotational bending test. As a result, a blurred surface of fatigue cracking is obtained with a relatively distinct zone of accidental fatigue in the core of the sample with a large deformation. Samples that break immediately, e.g. as a result of faulty workmanship (usually lack of straightness - very few cases) can be documented misleading the reader. They are not representative. In my opinion, the reviewer recommends discussing the impact of impurities on the formation and propagation of fatigue cracks based on crack micrographs for brittle materials. I don't know the method how to do it. Conclusion on the basis of an ad hoc breakthrough in the core of a sample with a high degree of deformation does not make sense in my opinion. For over 20 years of managing the work of the laboratory on an industrial scale, I have not been able to learn such an effective method. If I have a problem with this, I'd love to know the solution.
As a side note, I would like to add that there is interest in this type of work also in industry. The industry closely observes experimental work carried out in real conditions. Works based solely on tests on alloys made in the laboratory usually do not work in industry (the problem of scale). From the exchange of information, my results for steel containing mainly fine impurities are also confirmed by other steelworks.
- The conclusion should summarize the main findings and clarify how the results presented in the paper have made the field progressing. Future perspectives and research directions should also be included. A sole summary of the findings is not sufficient. Please polish the conclusion along this view.
Conclusions completed. The proposal for further research in the reviewed manuscript was presented in the last sentence of the Conclusions.
- This paper is like a statistical report, and many key results are not given. It need to be seriously revised.
I don't know what key results the reviewer has in mind. I agree with the reviewer that the presented manuscript is based mainly on statistical considerations. The occurrence of non-metallic inclusions in steel is random and cannot be considered otherwise than by means of tools designed to consider random distributions, such as statistics. Similarly, the results of fatigue strength tests based on a random variable are presented. If in other works the authors present the results of fatigue tests carried out on a laboratory scale on several samples made of hard steel prone to cleavage fatigue fracture, then they can analyze the phenomena occurring in micro-areas. In highly ductile steels, this is impossible. That is why the trial series is so numerous, too. Of course, the number of trials could be used for the analysis, in the presented manuscript a less favorable variant of the analysis of the arithmetic mean value of the trials was selected, and then each of the averages was adopted as a parameter describing the variable. Such a procedure significantly reduced the statistical significance of the research results, which I believe is still at a high level (correlation coefficients). Other research can be done (the question is whether they are needed to achieve a given goal or are to beautify the work), there is no work that cannot be improved yet. If the reviewer knows the ideal way to develop statistical results without using statistical parameters, please give me some guidance.
The revised manuscript has undergone external linguistic proofreading.
With the best regards
Tomasz Lipiński
Reviewer 2 Report
This manuscript presents results about the impact of size and distribution of impurities in a low carbon steel on its bending fatigue strength. Although this is an interesting topic and has industrial applications, the reviewer believes that the manuscript needs to be reworked. Some sections require more explanations and details. Therefore, the reviewer suggests that the manuscript requires Major Revisions.
1. Abstract
1) This section should be short and succinct and only highlight the main points about methodology and findings. Most of this section has been written in a way that is similar to literature review. Please revise.
2. Introduction
1) A comprehensive literature review has been conducted. However, it is not clear what is novel about this research. Please clearly explain in the manuscript what is it that this manuscript has managed to accomplish that was not offered by other studies.
2) The impact of the material on these results was not explained. Has the author conducted this study on other materials before?
3) Please explain what is specific about the material that was chosen for this study.
3. Materials and Methods
1) Are equations 3 and 4 derived by the author? Are they innovative equations used only in this study for calculation of distribution of impurities?
2) The methodology explained seems to be basic. Please explain if there was any element of novelty associated with the methodology in this study.
4. Results
1) The study of materials microstructure and discussion of impurities without SEM images seems to be not complete. Please include SEM images of the base material and highlight some impurities.
2) How were impurities diameters classified for irregular shapes? Is the classification conducted based on minimum or maximum diameter values? How does this affect the accuracy of results?
3) In Figures 1 – 11 there is no mention of error bars. This is not possible because of uncertainty associated with the classification of impurities size. Please explain.
4) In case the material is changed, the graphs shown in Figs. 2 -11 will be changed. It is not clear whether the trend is important, or the actual values obtained for this specific material. Please explain.
5) In measurement of impurities, how the inclusions were distinguished from oxides?
5. Conclusions
1) Please highlight the main finding of this study that was associated with the element of novelty.
The reviewer believes that the manuscript can be improved by being proofread by a native English speaker.
Author Response
Dear Reviver,
Thank you for your constructive review
- Abstract
1) This section should be short and succinct and only highlight the main points about methodology and findings. Most of this section has been written in a way that is similar to literature review. Please revise.
Abstract changed.
- Introduction
1) A comprehensive literature review has been conducted. However, it is not clear what is novel about this research. Please clearly explain in the manuscript what is it that this manuscript has managed to accomplish that was not offered by other studies.
In the Introduction, the reader's attention was drawn to the issues discussed in the manuscript that are important in the author's opinion. Conclusions have also been added.
2) The impact of the material on these results was not explained. Has the author conducted this study on other materials before? 3) Please explain what is specific about the material that was chosen for this study.
I conducted research on other materials, but belonging to the same group of steels. I haven't published their results yet, but they are consistent with the presented results. The main reason for the selection of the material for the study was the lack of research results in the literature on the impact of non-metallic inclusions on the fatigue strength of steel produced under the conditions carried out. Due to over 20 years of my industrial experience, when presenting, among other things, various aspects of research, I always relate them to industry. Tests conducted on a laboratory scale can serve as industry insights, but they cannot be directly transferred to industrial conditions. Mostly they are not confirmed in the industry. For this reason, voices can be heard coming from the industry that science is for itself and practice is for itself. Why highly ductile steel. If we analyze the research material used to determine the fatigue strength, it is not difficult to notice that researchers are very eager to reach for hard steels with a cleavable nature of the fatigue fracture. Such material is an easy and rewarding material for both research and presentation. It is easy to visualize the place of crack initiation and the direction of its development. In ductile materials, this is impossible. Steel with high ductility deforms, and the resulting micro-crack develops slowly, which results in mechanical scuffing of the place of discontinuity. In well-conducted tests that can be considered representative, it is not possible to show the fractography of the crack initiation site. It will always be obliterated (especially during rotational bending) as a result of the movement of both planes created as a result of steel decohesion. It is reassuring that in the last few years few papers on plastic steels have started to appear. Unfortunately, these are usually tests on a laboratory scale, which I understand considering the costs.
- Materials and Methods
1) Are equations 3 and 4 derived by the author? Are they innovative equations used only in this study for calculation of distribution of impurities?
Equations (3) and (4) are known to me from elementary school as linear equations. Probably the Reviewer made a mistake reading the numbering of the equations, and the question was about equations (2) and (3). Reference [75] is given for equation (3) in the manuscript, so it is not my equation. This is an equation derived by Sumit. It has been verified many times and is the canon of impurity steorology, similar to the Jonson and Hunt equations for growth on the crystallization front. Whereas [2] is a dependency developed under my direction.
2) The methodology explained seems to be basic. Please explain if there was any element of novelty associated with the methodology in this study.
Typical proven research methodology was used with the exception of (2).
- Results
1) The study of materials microstructure and discussion of impurities without SEM images seems to be not complete. Please include SEM images of the base material and highlight some impurities.
Steel microstructures for each tempering temperature and examples of non-metallic inclusions have been added.
2) How were impurities diameters classified for irregular shapes? Is the classification conducted based on minimum or maximum diameter values? How does this affect the accuracy of results?
In fact, the equivalent diameter of non-metallic inclusions was determined on the basis of their surface area. I have not emphasized this in the work due to the difficulties with understanding this issue by the reading room without a thorough knowledge of stereometry. Applying the law of transition from surface to volume is quite difficult. The diameter, on the other hand, is understandable to everyone.
Based on the test results, it was found that the largest number of inclusions had a diameter of less than 2 um. The remaining inclusions are quantitatively small in relation to this smallest fraction. This amount decreases exponentially as the impurity diameter increases. For example, a projection of a contamination with a diameter of 5 µm occupies an area of 19.63 µm and an inclusion with a diameter of 2 µm has an area of 3.14 µm, so the quotient is 6.25. For larger impurities the proportion is even higher. So this amount of big impurities is not that big. The study did not provide all the details of the study. It is not possible. Inclusions above 10 µm were found, but their distribution is always random, generally the larger the impurities, the less regular their distribution was. This, of course, results from the ratio of the observation area to the amount of pollution (scale effect). It is a statistical process. The research problem is quite significant and consists in the selection of magnification and observation area. The study used the observation area recommended by the standard, i.e. at 100x magnification, but for a more accurate analysis, leaving the same observation area, the magnification was increased to 400x. This was possible by assembling the image from several smaller images. This is a typical process for automated microscopic analysis. Going into such details would make this work very difficult to read and even for people who know image analysis quite complex. The analysis in this work is presented after analytical separation of large-size inclusions on the image analysis stand. Statistical tests with a very large number of samples that have been tested and the number of repetitions for each sample confirm the significance of such an analysis for diameters from 2.15 µm. The difference between a diameter of 2 µm and less than 2.15 µm is the analysis error region. As for the image analysis, this is a good result. The research problem remained: should the eliminated impurities with a diameter above 2 µm be replaced with impurities with a diameter of 2 µm or not? The difference in the coefficient of determination was around 0.05. After the analysis, it was decided that the larger pollutants eliminated were not replaced with others with a maximum analyzed diameter of 2 µm. There were many such problems and it is impossible to describe them all. The exact description of the analysis and its justification alone would be the volume of the work and maybe someday at some stereometric conference I will be tempted to do so. As a side note, I would like to add that there is interest in this type of work also in industry. The industry closely observes experimental work carried out in real conditions. The exchange of information shows that my results regarding the morphology of inclusions are also confirmed by other production plants.
3) In Figures 1 – 11 there is no mention of error bars. This is not possible because of uncertainty associated with the classification of impurities size. Please explain.
I partially answered above. Analysis of inclusion morphology is not easy. The analysis was carried out by two methods. A typical analysis on a computer station (and unfortunately the losses were as indicated by the reviewer), but also the pollutants themselves after chemical extraction were analyzed. Based on the results from this analysis, the impurity diameters were determined using high magnification. Unfortunately, the error of such an analysis could be determined only in relation to the volumetric analysis of inclusions and was usually at a negligible level. Can you do otherwise? Unfortunately, looking historically, the research results in each period of their conduct are burdened with the error of the research technique (access to the appropriate class of equipment), which depend on the development of technology (e.g. formerly the 1 m standard was made of iridium near Paris, and now it is described using the atomic method). We cannot avoid mistakes, but we must be aware of them. In Fig. 3 (former Fig. 1) error bars have been introduced. In the remaining Figs, there is no need as the regression coefficient is specified. It also gives information about the tolerance of the results. The distribution of non-metallic inclusions in industrially produced steel is random. Therefore, all the features associated with it are random.That is why the trial series is so numerous, too. Of course, the number of trials could be used for the analysis, in the presented manuscript a less favorable variant of the analysis of the arithmetic mean value of the trials was selected, and then each of the averages was adopted as a parameter describing the variable. Such a procedure significantly reduced the statistical significance of the research results, which I believe is still at a high level (correlation coefficients).
4) In case the material is changed, the graphs shown in Figs. 2 -11 will be changed. It is not clear whether the trend is important, or the actual values obtained for this specific material. Please explain.
There are several factors to consider when changing material, the most important being:
Are we changing the manufacturing process?
Do we change material to material from the same group? Or do we change dramatically, e.g. to bearing steel.
If we use the same production process and material from the same group, the results should not change much. On the other hand, a change in the manufacturing process or a major change in the group of material must cause a change in its properties. Such a question can be posed to any research result, and the answer to it lies within the scope of academic discussion. In the work, it is very important that with an increase in the number of small non-metallic inclusions (for steel of high quality - i.e. purity), the fatigue strength not only does not decrease, but even increases. This is confirmed not only in my industrial research for this grade of steel, but also for other grades and in other steelworks. Summing up, the significance of the presented relations applies only to steels with high plasticity and high purity.
5) In measurement of impurities, how the inclusions were distinguished from oxides?
Not distinguished. Only size was taken into account.
- Conclusions
1) Please highlight the main finding of this study that was associated with the element of novelty.
Highlighted.
The revised manuscript has undergone external linguistic proofreading.
With the best regards
Tomasz Lipiński
Round 2
Reviewer 1 Report
I carefully re-evaluated the revised manuscript and the author's response to the reviewer's comments. The authors have addressed all the concerns raised by and manuscript quality has been increased by the inclusion of suggested studies/results. I recommend the publication of the manuscript in its current format.
Author Response
Dear Reviver,
Thank you for your nice review.
I carefully re-evaluated the revised manuscript and the author's response to the reviewer's comments. The authors have addressed all the concerns raised by and manuscript quality has been increased by the inclusion of suggested studies/results. I recommend the publication of the manuscript in its current format.
With the best regards
Tomasz Lipiński
Reviewer 2 Report
The author has provided explanations about all the questions and has implemented some of the recommended changes. Although the manuscript has improved nicely, the reviewer believes that it is not ready for publication yet.
1) First, the title of the manuscript is too general. More details should be provided to narrow down the scope.
2) The most important factor in a research article is the novelty. This element is missing in the current form of the manuscript. The reviewer did not find any element of novelty in the material, methodology, analysis, or findings. There are MANY articles that has studied the impact of inclusions on the mechanical properties of steels including fatigue strength.
3) Thanks for adding Figures 1 and 2 to the manuscript. However, the difference between images in Fig. 1 is not explained. Please show the features that are of interest in the images. Also, please explain the importance of the different patterns shown in Fig. 2.
4) It is written that "the pollution distribution index is obtained as the quotient of the average size of pollutants and the average distance between pollutants". Please explain how much uncertainty is associated with this analysis. If you obtain the distance between the inclusions from SEM images, you will see that the pattern and the size of the inclusions would be different from one image to the other. If we consider the inclusions as spheres in a simplified case, the size of the sectioned 2D inclusion diameter would be dependent on the cut section. Please explain in the manuscript how much this fact can affect the results.
5) Most of the statements in the conclusions section are presented by using relative words. Please describe the results by using quantitative values rather than using qualitative terms such as low or high.
Author Response
Dear Reviver,
Thank you for your next review
- First, the title of the manuscript is too general. More details should be provided to narrow down the scope.
The title has been changed at the request of the reviewer.
2) The most important factor in a research article is the novelty. This element is missing in the current form of the manuscript. The reviewer did not find any element of novelty in the material, methodology, analysis, or findings. There are MANY articles that has studied the impact of inclusions on the mechanical properties of steels including fatigue strength.
The novelty of the manuscript is presented in the last two paragraphs of the Introductions. I'm sorry the reviewer can't see what's new. Apparently, there are works by other authors linking the diameters of non-metallic inclusions with the distance between the precipitates. I don't know them. There are probably works developed on the basis of research on an industrial scale describing the fatigue strength of high-quality low-carbon steel as a dependence on the volume of non-metallic inclusions (with similar impurity morphology) presented by other authors, I would love to read them. In my opinion, the results of full tests carried out on an industrial scale (which I have presented) cannot be compared with simplified tests carried out on a laboratory scale, the results of which can only be treated as pilot tests for proper research. How many items of literature present research on an industrial scale? There are probably studies by other authors that present similarly described changes to those presented by me. However, I can't find these. Please, show me such works, and thus confirm the correctness of your views. I will be happy to enrich my knowledge.
After all, one may wonder if the topic presented in my manuscript is completely exhausted, and the paper does not contribute anything to the research problem? Do the available works of other authors fully explain the issue? Then further research may be unjustified. I am unable to point to a corroborating justification for the above questions, so I cannot make such a statement, much less answer it.
3) Thanks for adding Figures 1 and 2 to the manuscript. However, the difference between images in Fig. 1 is not explained. Please show the features that are of interest in the images. Also, please explain the importance of the different patterns shown in Fig. 2.
I supplemented the entry in the manuscript, also provided references to sample literature {78-80].
4) It is written that "the pollution distribution index is obtained as the quotient of the average size of pollutants and the average distance between pollutants". Please explain how much uncertainty is associated with this analysis. If you obtain the distance between the inclusions from SEM images, you will see that the pattern and the size of the inclusions would be different from one image to the other. If we consider the inclusions as spheres in a simplified case, the size of the sectioned 2D inclusion diameter would be dependent on the cut section. Please explain in the manuscript how much this fact can affect the results.
Error record added.
The diameter of the impurities was transposed to the volume in accordance with the Apis given in the paper based on the known physical relationship. It was confirmed by testing non-metallic inclusions obtained by chemical extraction from a unit volume. Since the error (for a large number of measurements) was lower than 5% (and such a statistical level of confidence was assumed in the work), the computer method was further used. If the error was large, the volume of non-metallic inclusions would be determined by chemical extraction and then particle size analysis also using a computer system. I emphasize that (according to the entry in the manuscript) the magnification of 400x was used for the analysis of inclusions, while maintaining the field of observation required by the standards of 100x. Thus, the error was minimized (with the required field of view for 100x, magnification of 400x gave much smaller errors). It should also be noted the entry in the manuscript according to which the distance between the inclusions was not measured, but calculated according to equation (2), as I already explained to the Reviewer in response to previous comments. I also wrote there that the substitute diameter was determined. The ranges of the minimum size of the diameters counted using the computer technique were defined in the image analysis program. Determination of the surface of impurities smaller than 2 µm (generally smaller than 1 µm) was burdened with a relatively large error (although up to the size of 1 µm it was within the range adequate within the confidence interval for α=0.05). Therefore, total impurities were estimated by chemical extraction. The dimensional range with a diameter of less than 2 µm was calculated as the difference of all impurities and impurities greater than or equal to 2 µm (from the computer method).
It should be noted that the analysis is based on statistical parameters. Thus, one cannot measure any distance between inclusions and say that the presented analysis is inadequate. Take a suitable sample, make a certain number of measurements for it (not the distance between two inclusions) and then estimate the average, and I am confident that the result will coincide with the results presented in the manuscript. It will converge because that was the methodology used. However, it was verified by determining the volume of impurities in a unit volume of steel in many samples. On another steel, verification tests were carried out with a significant narrowing of the measured diameters and the results were comparable (slightly more accurate, resulting from smaller differences in the size of inclusions in individual ranges). The general law cannot be cheated. The tests were also carried out independently by various manufacturers of measuring equipment (who wanted to sell their solution). Please believe that regardless of the offered systems and manufacturers, the results were comparable. This is confirmed by the fact that at the existing level of technology, all equipment vendors offer similar technological solutions. In order to overcome the existing error, we must wait for the so-called technical leap.
The estimate of the error in the manuscript was given by estimating the correlation coefficient of the presented equations (deviations of the measurement points from the regression lines describing their distribution).
5) Most of the statements in the conclusions section are presented by using relative words. Please describe the results by using quantitative values rather than using qualitative terms such as low or high.
In part, the text was supplemented with specific values. However, for fear that the constant repetition of the values once given (in a given paragraph) may also be criticized by the reviewer, some terms called relative by the reviewer were left unchanged. It should be noted that also available literature, standards and guidelines for the production and evaluation of materials use such terms, e.g. high quality steel, high purity steel, low carbon steel, low impurity steel, low carbon steel, etc. .high yield steel. If we assume ductility as high at 8%, and steel will have 7.9%, will it already be of medium or low ductility? In addition, the plasticity assumed at the conventional 8% may apply to a specific steel after a specific heat treatment, etc. In the literature, it is assumed that plastic steels are simply called steels capable of plastic deformation, and this does not always go hand in hand with their elongation. I do not know how to parameterize this concept and, as far as I know, it is not parameterized in the literature either. This also applies to many other terms used.
If still so-called relative terms are glaring, please indicate them and I will correct them by duplicating quantitative entries.
In the current review, the reviewer has raised several issues not mentioned in the first review. Some of the comments have the character of an academic discussion. I hope that I have answered the questions posed to me. If the reviewer comes up with further new comments, I will gladly respond to them in the next answer, if only the editor would allow me to do so.
With the best regards
Tomasz Lipiński